# From the Beehives: Identification and Comparison of Physicochemical Properties of Algerian Honey

**Mokhtar Guerzou** [1] , **Hani Amir Aouissi** [1] , **Ahlem Guerzou** [2] , **Juris Burlakovs** [3] , **Salaheddine Doumandji** [4] **and Andrey E. Krauklis** [5,*]

1   Scientific and Technical Research Center on Arid Regions (CRSTRA), Biskra 07000, Algeria; gmfmokh@gmail.com (M.G.); aouissi.amir@gmail.com (H.A.A.)
2   Laboratory of Exploration and Valorization of Steppe Ecosystems, Faculty of Natural and Life Sciences, University Ziane Achour, Djelfa 17000, Algeria; dreamsdj@yahoo.fr
3   Rural Building and Water Management, Estonian University of Life Sciences, Kreutzwaldi 5, 51014 Tartu, Estonia; Juris.Burlakovs@emu.ee
4   Laboratory of Zoology, National Superior School of Agronomy El Harrach, Algiers 16000, Algeria; s.doumandji@ensa.dz
5   Institute for Mechanics of Materials, University of Latvia, Jelgavas Street 3, LV-1004 Riga, Latvia
*   Correspondence: andykrauklis@gmail.com

**Abstract:** In this study, the authors aimed at characterizing 11 Algerian kinds of honey taken from various geographical locations (beehives located at Djelfa (Medjbara and Dzaira), Laghouat, Aflou, Medea, Tiaret, Sidi bel-Abbes, Tiaret, Ain-Safra, Mostaganem, El Bayadh, and Ghardïa). The authors investigated the physicochemical parameters of these honey samples, including density, water content, electrical conductivity, ash content, pH, hydroxymethylfurfural (*HMF*) content, free acidity, and color. The physicochemical parameters obtained were found to be within acceptable ranges according to the international standards (Codex Alimentarius) for 9 out of 11 analyzed samples: density 1.38–1.50 g/cm$^3$ (the same as kg/L, as commonly used for honey), water content 14.03–18.80%, electrical conductivity $0.38 \times 10^{-1} - 6.41 \times 10^{-1}$ mS/cm, ash content 0.06–0.48%, pH 3.50–4.50, free acidity 11–47 meq/kg, and color 1.1–9.2 Pfund index. Analysis of *HMF* content showed that only two honey samples have high values (117.7 and 90.0 meq/kg). Most samples of Algerian honey are of suitable quality according to international standards.

**Keywords:** quality; food; physicochemical parameters; physicochemical analysis; honey; hydroxymethylfurfural; Algeria

## 1. Introduction

Honey is a natural sweet substance produced by honey bees from the nectar of plants or from secretions of living parts of plants, or excretions of plant-sucking insects on the living parts of plants, which the bees collect, transform by combining with specific substances of their own, deposit, dehydrate, store and leave in the honeycomb to ripen and mature [1]. In other words, honey is a natural, complex food and nutritional commodity prepared by honeybees.

Honey is known to be a complex mixture of sugars (carbohydrates such as monosaccharides, glucose, and fructose) and other components [2], such as proteins, minerals, vitamins, organic acids, and enzymes produced by honey bee workers [3,4]. In general, the chemical composition of honey has been shown to be affected by many factors, which therefore makes it an interesting subject of study [5–7]. In addition, honey is considered a medicinal food with antioxidative, anti-inflammatory, and antibacterial properties [8]. Therefore, monitoring honey quality characteristics is of great importance to consumer health [9,10].

The quality of honey depends on moisture content, temperature, sugar content, botanical origin, and geographical location and can be characterized by its physiochemical

properties, which were investigated in this study. Several physicochemical parameters have been used in the characterization and quality control of honey worldwide [11–14]. Not to mention the studies that have focused on the quality of honey from certain countries such as: Japan [15], Pakistan [16], Zambia [17], China [18], and Spain [19,20] as examples.

Undoubtedly, some studies have been carried out on honey in Algeria [21–31]. This was also the case for neighboring countries such as Morocco and Tunisia [32,33]. However, in view of the importance of this substance, the characteristics remain relatively unknown, and by comparison with studies carried out around the world, the number of articles remains insufficient. In general, Algeria remains understudied in the field of natural sciences, as reported in a recent study [34].

Among the recent studies that have been conducted, Zantaz honey (rich in methyl syringate) was investigated in detail in an article, which subsequently led to an increase in its market value and economic benefits [35,36]. Zerrouk et al. (2017) [24] focused on the characterization of jujube (*Ziziphus lotus* (L.) Lam). Later, a second study focused on qualitative pollen analysis and showed the participation of nectarless plants in the formation of honey (note that nectarless plants do not participate in the formation of honey per se, but their pollen grains may be present in honeys because they are unintentionally brought to the hive by bees on their hairs or arrive in nectar and honeydew as they are carried by the wind), specifically *Chenopodium* sp., and *Olea europaea* [37].

Considering the intent of a consumer to find natural products for daily life, including beehive products, this study investigates and compares the quality of honey products. The work sites were carefully selected to place honey boxes, with natural sites that were difficult to reach for the average person.

Furthermore, to highlight the practical importance of this study, it should be noted that national honey production is currently estimated to be 33,000 kg, with a yield of 4 to 8 kg per hive. This is considered very low compared to the honey potential offered by Algeria, as the country imports more than 150,000 tons of honey per year from China, India, Saudi Arabia, and other countries [38]. Therefore, studying and reporting the physicochemical characteristics of Algerian honey samples is of high significance.

## 2. Materials and Methods

### 2.1. Honey Samples

To achieve the defined goal, the authors worked on eleven samples of honey that were brought from hives located at Djelfa (Medjbara and Dzaira), Laghouat, Aflou, Medea, Tiaret, Sidi bel-Abbes, Tiaret, Ain-Safra, Mostaganem, El Bayadh, and Ghardïa (Figure 1). Samples were taken in each apiary in each region. The sample represents the apiary, and the same conditions (e.g., amount of honey, same bees, etc.) were applied to every single sample. We have intentionally opted for honeys from the most famous regions in the production of honey in order to have a global sample that is representative of what is found at the national level. Based on past studies [39,40], it appeared that our sample was sufficient to provide an overview of the honeys of the whole country.

The honey samples were collected between 2018 and 2019 and stored at 4–6 °C in the absence of light. Botanical classification was achieved in cases where the pollen spectrum contained more than 45% of the respective dominant pollen [41,42]. Pollen types were identified by comparing them with a reference collection from the laboratory of analysis of the honey located at Baba Ali (Blida, Algeria) (Table 1).

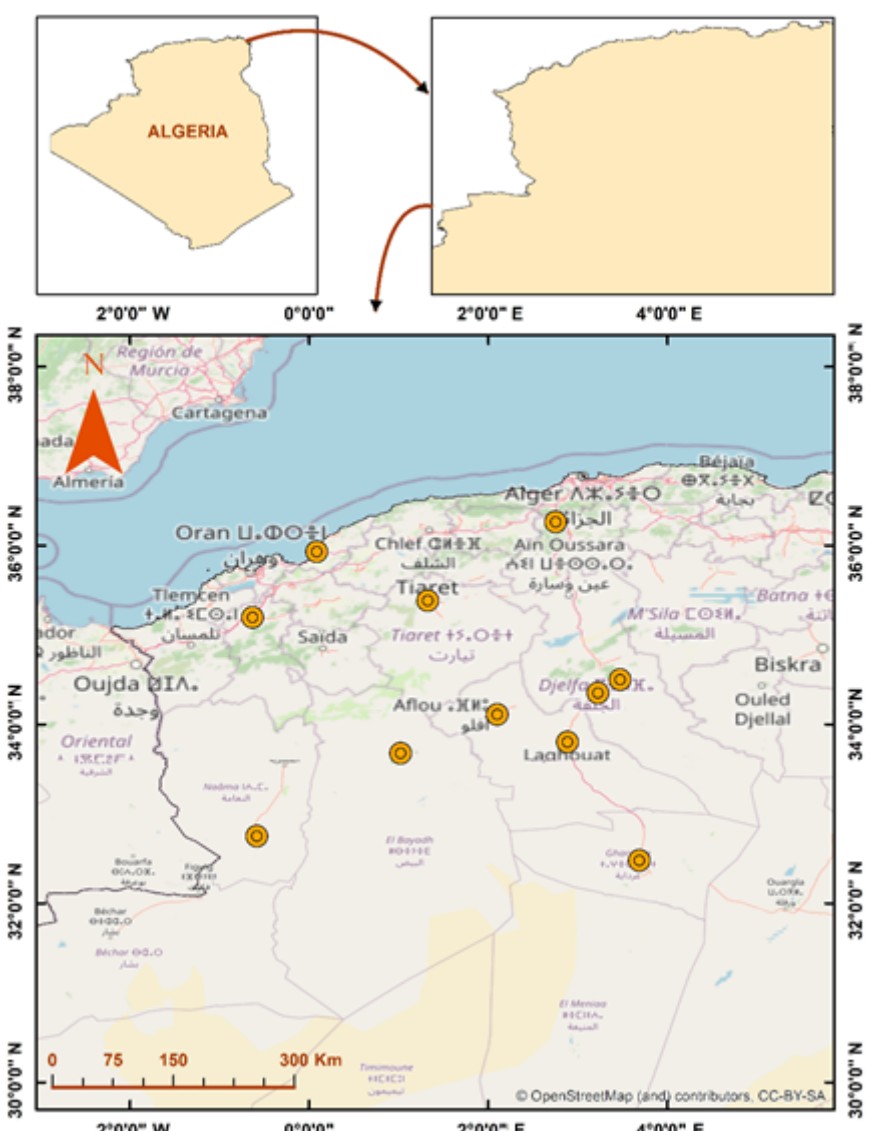

**Figure 1.** Geographic area and location of collected honey samples.

**Table 1.** Honey samples and their botanical origin.

| N° of Sample | Harvest Period | Location | Denomination | Type of Extraction |
|---|---|---|---|---|
| H01 | June 2018 | Djelfa (Dzairia) | Multifloral | Manual |
| H02 | June 2018 | Djelfa (Moudjebara) | Multifloral | Manual |
| H03 | June 2018 | Tiaret | Multifloral | Mechanical |
| H04 | June 2018 | Medea | Aleppo Pine | Mechanical |
| H05 | June 2018 | Aflou | Multifloral | Mechanical |
| H06 | June 2018 | Laghouat | *Ziziphus lotus* | Mechanical |
| H07 | May 2019 | El Bayadh | Multifloral | Mechanical |
| H08 | July 2019 | Mostaganem | *Eucalyptus* | Mechanical |
| H09 | August 2019 | Ain-Safra | *Ziziphus lotus* | Mechanical |
| H10 | June 2019 | Sidi Bel-Abbes | Multifloral | Mechanical |
| H11 | June 2019 | Ghardaia | *Phoenix sp.* | Manual |

Note: H: honey.

### 2.2. Physicochemical Characterization

#### 2.2.1. pH and Free Acidity

The pH was measured using a LIDA220 pH-meter (Tsingtao Unicom-Optics Instruments Co., Ltd., Shanghai, China) in a solution containing 10 g of honey in 75 mL of distilled water. The free acidity was obtained by plotting the neutralization curve with a NaOH

solution and by determining the acidity (pH) of the equivalence point (pHe). The acidity of the lactones was obtained by adding excess NaOH to the honey solution, following the method of Bogdanov et al. (1999) [43], and plotting the curve of neutralization for the excess sodium hydroxide by back titration with sulfuric acid. The results are expressed as meq/kg [44].

### 2.2.2. Density

The density of a substance is defined as its weight per unit volume; specific gravity is the ratio of the weight of a substance to the weight of the same volume of water at a specified temperature [45].

### 2.2.3. Electrical Conductivity

The electrical conductivity of honey is defined as that of 20% weight-in-volume solution in water at a 20 °C temperature; the 20% refers to the dry matter of honey. Results are expressed in milliSiemens per centimeter (mS/cm) [46], according to the unit of measurement (mS/cm) reported in Codex Alimentarius and harmonized methods of the European Honey Commission.

### 2.2.4. Ash Content

The ash content of honey is defined as the residue obtained by a well-defined procedure and expressed as a percentage by weight. Preparation of the ash involved heating the dish in the electric oven at the ashing temperature, followed by cooling in a desiccator back to room temperature and weighing to 0.001 g precision ($m_2$). Weighing the 5–10 g of the sample (to the nearest 0.001 g) into an ash dish, as described above, this was previously prepared ($m_0$). Two drops of olive oil were added. Then, water was removed, and the ashing procedure was started without loss at a low heat rising up to 350–400 °C by using one of the appliances. After the preliminary ashing, the dish was placed in the preheated furnace and heated for at least 1 h. Subsequently, the ash dish was cooled in the desiccators and weighed. The ashing procedure was continued until a constant weight was reached ($m_1$). The ash proportion ($W_A$) in g/100 g honey was calculated via the following equation [43]:

$$W_A = \frac{m_1 - m_2}{m_0} \times 100 \tag{1}$$

where $m_0$ = weight of the honey; $m_1$ = weight of the dish and ash; $m_2$ = weight of the dish.

### 2.2.5. Moisture Content

Water values were determined using an ATAGO NAR-3T Abbe refractometer at 20 °C (Fisher Scientific, Waltham, MA, USA). The moisture content in the investigated honey samples was determined. It was between 14.03% and 18.8%, well within the limit ($\leq$20%) recommended by the international quality regulations (Codex Alimentarius Commission) (Table 1). Water content is of high importance for the shelf life of honey during storage and, if too high, can lead to undesirable honey fermentation due to osmotolerant yeasts, which form ethyl alcohol and carbon dioxide [47]. Generally, all of the investigated Algerian honey samples in this study were of suitable quality, as was previously demonstrated in one of the Algerian studies by Khalil et al. (2012) [48]. This result is indicated by the low moisture content.

### 2.2.6. Color Measurement

The color was measured by the technique of a visual Lovibond comparator [49]. The liquid honey was loaded into the measuring tube, the color was compared with standards, and the results obtained were expressed as "Pfund index".

### 2.2.7. Hydroxymethylfurfural Analysis

The presence in the honey of hydroxymethylfurfural (2-hydroxymethyl-5-furaldehyde, *HMF*) was originally considered evidence of its adulteration with acid-converted invert

syrup [50]. An organic compound known as 5-hydroxymethylfurfural (*HMF*) is formed from reducing sugars in honey and various processed foods in acidic environments when they are heated through the Maillard reaction. In addition to processing, storage conditions affect the formation of *HMF*, and *HMF* has become a suitable indicator of honey quality [51]. About 5 g of honey (weighed to 1 mg precision in a small beaker) was transferred to a 50 mL volumetric flask with a total of 25 mL distilled water. Then, 0.50 mL of Carrez solution I was added, followed by mixing and adding 0.50 mL of Carrez solution II. Then, the solution was well mixed and diluted with water to a specific volume. A drop of alcohol was added in order to suppress surface foaming. Subsequently, filtering through paper was performed, rejecting the first 10 mL of the filtrate. A total of 5 mL of the filtrate were pipetted into each of the two $18 \times 150$ mm test tubes; 5 mL of water was pipetted into one (sample), and 5 mL 0.20% bisulfite into the other (reference). The solution was well mixed using a vortex mixer, and the absorbance of the sample was determined against the reference in 1 cm cells at 284 and 336 nm. In cases when absorbance was too high for accuracy (>0.6), the sample solution was diluted with water as needed (and reference solution to the same extent with 0.1% $NaHSO_3$). Then, absorbance values need to be multiplied by an appropriate dilution factor before the calculation [52]. The *HMF* in mg/100 g honey is calculated using the following formula (in mg *HMF*/100 g honey):

$$HMF = (A_{284} - A_{336}) \times Factor \times \frac{5}{sample} \tag{2}$$

where $A_{284}$ = Absorbance of *HMF* at 284 nm; $A_{336}$ = Absorbance of *HMF* at 336 nm; *Factor* = 14.97.

### 2.2.8. Statistical Analysis

All results were statistically analyzed via the one-way analysis of variance (ANOVA). The difference was considered significant for ($p < 0.05$).

### 3. Results

The results of the physicochemical characteristics of honey samples are presented in Table 2. All parameters varied greatly in almost all cases ($p < 0.001$).

**Table 2.** Analysis of some physicochemical parameters of *Apis mellifera* L. honey samples.

| Samples Code | Density (g/cm$^3$) | Electrical Conductivity ($10^{-1}$ mS/cm) | pH | Water Content (%) | Ash Content (%) | HMF (mg/kg) | Free Acidity (meg/kg) | Color (Pfund Index) |
|---|---|---|---|---|---|---|---|---|
| H01 | 1.50 ± 0.02 | 1.12 ± 0.004 | 4.24 ± 0.01 | 14.8 ± 0.08 | 0.08 ± 0.001 | 2.84 ± 0.54 | 11.0 ± 0.01 | 1.1 |
| H02 | 1.38 ± 0.01 | 6.41 ± 0.005 | 4.5 ± 0.01 | 18.8 ± 0.05 | 0.48 ± 0.001 | 5.68 ± 0.02 | 11.5 ± 0.01 | 9.9 |
| H03 | 1.50 ± 0.06 | 2.34 ± 0.010 | 3.5 ± 0.10 | 14.03 ± 0.01 | 0.17 ± 0.003 | 117.7 ± 0.20 | 25.0 ± 0.50 | 14 |
| H04 | 1.45 ± 0.03 | 1.48 ± 0.004 | 3.7 ± 0.10 | 15.8 ± 0.05 | 0.11 ± 0.002 | 90.7 ± 0.50 | 47.0 ± 0.30 | 14 |
| H05 | 1.46 ± 0.01 | 1.46 ± 0.010 | 4.16 ± 0.02 | 14.4 ± 0.60 | 0.11 ± 0.001 | 9.28 ± 0.01 | 37.0 ± 0.01 | 6.2 |
| H06 | 1.45 ± 0.01 | 2.61 ± 0.010 | 4.25 ± 0.01 | 15.03 ± 0.01 | 0.19 ± 0.020 | 9.92 ± 0.02 | 15.0 ± 0.01 | 6.2 |
| H07 | 1.45 ± 0.01 | 0.83 ± 0.010 | 4.48 ± 0.01 | 6.2 ± 0.01 | 0.06 ± 0.001 | 3.83 ± 0.70 | 18.0 ± 0.01 | 1.1 |
| H08 | 1.42 ± 0.02 | 5.06 ± 0.001 | 4.42 ± 0.01 | 17.4 ± 0.15 | 0.38 ± 0.001 | 12.87 ± 0.10 | 20.0 ± 0.20 | 8.3 |
| H09 | 1.47 ± 0.03 | 1.6 ± 0.005 | 1.4 ± 0.01 | 14.6 ± 0.05 | 0.12 ± 0.001 | 4.19 ± 0.04 | 16.5 ± 0.02 | 6.2 |
| H10 | 1.44 ± 0.01 | 1.38 ± 0.002 | 4.16 ± 0.01 | 16.2 ± 0.20 | 0.1 ± 0.003 | 7.43 ± 0.20 | 11.0 ± 0.70 | 7.1 |
| H11 | 1.46 ± 0.09 | 3.49 ± 0.003 | 4.38 ± 0.01 | 15.1 ± 0.17 | 0.26 ± 0.001 | 12.25 ± 1.34 | 26.0±0.12 | 8.3 |

For the density, the values recorded for the honey tested are included in the range of Codex Alimentarius, with an average density (D = 1.38 to 1.5 g/cm$^3$). All samples studied were honey acidic in nature, and the pH varied between 3.50 and 4.50. The free acidity of analyzed honey samples was between 11.0 and 26.0 meq/kg.

The highest value for the water content has been found in the H02 sample (18.8%). The electrical conductivity was less than $8 \times 10^{-1}$ mS/cm. The electrical conductivity

found in all the samples was typical for floral honey. HMF in nine honey samples was less than the maximum allowed (40 mg/kg) recommended by the Codex Alimentarius [1].

The percentage of ash content is an indicator of the mineral content. It has always been considered as a quality criterion that indicates the eventual botanical origin of honey. Its value in the analyzed samples ranged from 0.06% to 0.48% (Table 2).

The color of honey varies from light to dark depending on Pfund values, which differ according to the botanical origin. The tested honey falls into two categories: clear (9 samples) and dark (2 samples). The intensity of the color is directly related to the botanical origin and physicochemical parameters tested (pH, electrical conductivity, HMF, and ash content).

## 4. Discussion

### 4.1. Density

Honey samples have different density values ranging from 1.38 to 1.50 g/cm$^3$, which mostly meet the criteria defined by Codex Alimentarius standards, which range from 1.39 to 1.52 g/cm$^3$. The lowest value is exhibited by sample H02, indicating a value of 1.38 $\pm$ 0.01 g/cm$^3$, being on the edge of the lower boundary (1.39 g/cm$^3$). Chefrour et al. (2009) [53] reported density values of northeast Algerian honeys similar to those reported in this study, ranging from 1.37 to 1.5 g/cm$^3$. Similarly, analyzing honey from Saudi Arabia, Al-Khalifa, and Al-Arify (1999) [54] reported lower values ranging between 1.42 and 1.40 g/cm$^3$.

### 4.2. Electrical Conductivity

The obtained results demonstrated a consistent variability of electrical conductivity of 11 honey samples tested. They vary from $0.83 \times 10^{-1}$ to $6.41 \times 10^{-1}$ mS/cm. Lower values are noted by other authors as Makhloufi et al. (2007) [23], who reported a value of $0.65 \times 10^{-1}$ mS/cm. Zerrouk et al. (2011) [55] found that the electrical conductivity of honey is $2.75 \times 10^{-1}$ mS/cm for local honeys of Djelfa and $7.19 \times 10^{-1}$ mS/cm for honeys of Laghouat. Chefrour and al. (2009) [53] reported that values vary from $2.01 \times 10^{-1}$ to $9.22 \times 10^{-1}$ mS/cm. According to the Codex Alimentarius (2001) [1], nectar honey should have a lower electrical conductivity of $8 \times 10^{-1}$ mS/cm, unlike honeydew, which should have electrical conductivity greater than $8 \times 10^{-1}$ mS/cm. The relationship between the honey's electrical conductivity and the concentration of organic acids, mineral salts, and proteins has always existed. It is a parameter that shows great variability according to the floral origin and is considered one of the most important parameters for differentiating between blossom honey and honeydews [56,57].

### 4.3. Ash Content

The ratio of the mineral content varied from 0.06% to 0.48%. Sample H07 from El Bayadh has the lowest value (0.06%). These values are well within the standard set by the Codex Alimentarius [1], which is $\leq$0.6% for nectar honey or honeydew honey mixed to nectar and $\leq$1% for the honeydew honey (Bogdanov et al. 1997). These results are also consistent with those reported by other authors in Algeria, for example, by Zerrouk et al. (2011) [55] and Ouchemoukh et al. (2007) [25], with values between 0.07% and 0.33%, and 0.06% and 0.54%, respectively. Abroad Al-Khalifa and Al-Arify (1999) [54] reported that the mineral content of honeys in Saudi Arabia ranges from 0.02% to 0.59%, Naman et al. (2005) [58] indicate that this content is 0.32% for Moroccan honey, Saxena et al. (2010) [59] report that the Spain honeys content ranges from 0.03% to 0.24%.

### 4.4. Water Content

The water content of the honey samples ranged from 14% to 18.8%. Except for sample H02 (18.8%), all honey analyzed have values consistent with the standard set by the Codex Alimentarius [1], not exceeding 18%. Makhloufi et al. (2009) [60] treated 22 samples from different regions of Algeria and measured values close to those reported in this

study, reporting values between 13.9% and 20.2%. Percentages, reported by Ajlouni and Sujirapinyoku (2010) [61], of Australian honeys, were between 10.6% and 17.8%. This parameter gives information on the state of conservation of honey. An increase in moisture content can lead to undesirable honey fermentation during the storage stage. According to Bogdanov et al. (1997) [47], this increase in moisture content is caused by the osmotolerant yeasts' action, which results in the formation of ethanol and $CO_2$. The ethyl alcohol can be further oxidized to acetic acid and water, resulting in a sour taste [62]. Furthermore, the moisture content of honey can depend on many factors, such as the degree of maturity reached in the hive, harvesting season, and climatic factors [63].

### 4.5. Hydroxymethylfurfural

*HMF* content of honey varies strongly for the studied samples, from 2.84 to 117.7 mg/kg, respectively. The authors have found that nine samples numbered (H01, H02, H05, H06, H07, H08, H09, H010, and H011) have sufficiently low values (2.84 < *HMF* < 21.25) mg/kg. These results match the Codex Alimentarius (≤40 mg/kg). Khalil et al. (2012) [48] have analyzed some types of Algerian honey and have shown that the *HMF* is in low concentrations and their honey samples are of suitable quality. Similarly, Soria et al. (2004) [64] analyzed honey from Spain and found a range of values of *HMF* from 0.00 to 15.65 mg/kg. In the present work, results indicate that samples numbered (H03 and H04) have the highest rate of *HMF*, fluctuating between 90.7 and 117.7 mg/kg. Mendes et al. (1998) [65] argue that for rates up to 145.5 mg/kg, this value may be a result of heat treatment. Bath and Singh (1999) [66] and Zappala et al. (2005) [67] argue that too much heat and poor storage conditions can increase *HMF*.

### 4.6. Color

The results for this parameter showed significant variations between 11 mm Pfund for the samples H01 and H07 and 140 mm Pfund for the sample H04. Lower values are indicated by Zerrouk et al. (2011) [55] for honey "Djelfa" ranging between 41 and 92 mm Pfund. It should be remembered that honey has various colors, from creamy white to almost black brown. According to Gonnet (1985) [68], the colorimetric Pfund scale measures the color with an index ranging from 1 (clear honey) to 14 (dark honey). All the kinds of honey analyzed in this study meet this level except for honey from Tiaret that was too dark, and was not commensurate with the scope of color Pfund. Gonzales et al. (1999) [69] argue that honeys change their original color after 90 days of storage at 37 °C. It is noteworthy that when handling, the authors have noticed that honey from Tiaret was too dark, indicating that this honey was likely overheated.

### 4.7. pH and Free Acidity

All of the tested Algerian honey samples were acidic in their nature; the pH measurement attested to this with a variation between 3.50 and 4.50. Honey is a naturally acidic substance, regardless of its geographical origin; it is known that the organic acids' presence contributes to its stability against microbial spoilage during the conservation and its flavor too. The pH of honey samples is important during the extraction process because it affects the texture of honey as well as its stability and shelf life [57]. Tested samples have values of free acidity ranging from 11.0 to 47.0 meq/kg. These values are comparable with some honey produced in Algeria, as reported by Chefrour et al. (2009) [53]. Free acidity was less than the maximum limit specified as satisfactory in the international trade (50 meq/kg of honey), which indicates the absence of undesirable fermentation in the investigated samples.

### 5. Conclusions

The physicochemical properties of 11 Algerian honeys were investigated and obtained in this work. The honey samples were taken and analyzed from various geographical locations (beehives located at Djelfa (Medjbara and Dzaira), Laghouat, Aflou, Medea, Tiaret,

Sidi bel-Abbes, Tiaret, Ain-Safra, Mostaganem, El Bayadh, and Ghardïa). Density, water content, electrical conductivity, ash content, pH, hydroxymethylfurfural (*HMF*) content, free acidity, and color were studied and reported.

It was found that the majority of investigated Algerian honey specimens were of suitable quality according to international standards. Respectively, 9 out of 11 analyzed samples have shown representative physicochemical parameters, which conform to the established standards by Codex Alimentarius. Analysis of *HMF* content showed that only two honey samples (H03 and H04 from Tiaret and Medea, respectively) had higher values (117.7 and 90.0 meq/kg) than in the standard.

This study can be of help in choosing better quality honey products regarding their physicochemical parameters. However, each individual's expectations, needs, and preferences vary widely. It was, therefore, logical to provide consumers with more information on the honeys available in Algeria. Additional studies in the characteristics of honey according to their color and trace elemental content related to the origin of honey are planned. All in all, considering the physicochemical properties, Algerian honey has proven to be of competitive value on a global level.

**Author Contributions:** Conceptualization, M.G., H.A.A. and A.E.K.; methodology, M.G. and A.G.; software, M.G. and A.G.; validation, M.G., H.A.A. and S.D.; formal analysis, M.G., H.A.A. and A.E.K.; investigation, M.G.; resources, S.D.; data curation, A.E.K. and J.B.; writing—original draft preparation, M.G., H.A.A. and A.E.K.; writing—review and editing, H.A.A., J.B. and A.E.K.; visualization, H.A.A., J.B. and A.E.K.; supervision, J.B., S.D., and A.E.K.; All authors have read and agreed to the published version of the manuscript.

**Funding:** This research was funded by the DGRSDT and MESRS (Algerian Ministry).

**Institutional Review Board Statement:** Not applicable.

**Informed Consent Statement:** Not applicable.

**Data Availability Statement:** The data presented in this study are collected within a research project and can be made available on request from the corresponding author.

**Acknowledgments:** Many thanks are addressed to the DGRSDT and MESRS for their support. Saifi Merdas helped with the location map. Andrey Krauklis is especially grateful to Oksana Golubova for her support.

**Conflicts of Interest:** The authors declare no conflict of interest. The funders had no role in the design of the study, in the collection, analyses, or interpretation of data, in the writing of the manuscript, or in the decision to publish the results.

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
