# Peer review of "From the Beehives: Identification and Comparison of Physicochemical Properties of Algerian Honey"

_resources, doi:10.3390/resources10100094_

Round 1
Reviewer 1 Report
The authors have a knowledge of the literature of the topic, however a lot of references is not properly applied. The manuscript appears not too original, providing simple information about Algerian honeys. The text needs some implementations and revisions, reported in my comment. I suggest to give greater significance and prominence to the results and improve the conclusion section also by adding add more relevant considerations and appropriate references

Author Response
We want to thank you for your precious time in reviewing our paper and also for providing valuable comments. Undoubtedly, it was your insightful comments that led to possible improvements in the current version. We have carefully considered the comments and tried our best to address every one of them. We appreciate the feedback, we tried to apply corrections, improve references and enhance the conclusion at the same time, as suggested. We hope the revised version meet your high standards.

Reviewer 2 Report
The work must be reviewed in terms of content, format and language. The file with the comments is attached.

Author Response
Thank you for all the constructive comments and suggestions that were fully taken into consideration to carry out the necessary improvements and corrections. It is an honor for us to benefit from your experience. We completely agree on the fact that the article presented some weaknesses, and we have done our best to increase the overall level of the manuscript, and especially to respect your suggestions and recommendations point by point, we hope that the efforts we have put in will suit you.

Reviewer 3 Report
The authors can improve the introduction. The last paragraph is quite unrelated to the title of the manuscript. Of course, the authors can take this decision.
Material and methods: It is clear that there is only one honey sample from each area (region). Is this sample collected from an apiary or from one hive from the apiary?
The authors can collect more honey samples. For example they can collect honey samples from three bee hives from each apiary or more.
Are these apiaries registered? Do they have any legal organization which oversees them?
The authors can include information on how much honey Algerian beekeeping exports or imports to the world market? How much honey is consumed per capita in the country for 1 year?
The manuscript is interesting due to the fact that it is the first report for this area. But as a first report, it was good to be more extensive in terms of studied parameters and larger number of honey samples.
Probably, the financial support had an impact for this study.
Technical errors found:
Line 52: the characteristics remain relatively unknown.
Line 111: Weighing the 5-10 g of the sample
Line 113: Two drops of olive oil were added.
Line 238: maybe “their honey samples are of good quality”.
Author Response
First of all, we appreciate the time and effort that you have dedicated on providing your valuable feedback on this manuscript. We are very grateful for the insightful comments on our article.
Then, we agree that it is an excellent suggestion to improve the introduction. Indeed, the last paragraph was very loosely linked to the main theme of the study. Therefore, based on your suggestion, we corrected this and improved the introduction by adding instead, a part that talks about the importance of this study.

Reviewer 4 Report
The introduction is insufficient. The authors must add information why this study is important.
The last part of the introduction “Other rare studies have focused on propolis” is very far from the main goal of the study.
Materials and Methods: Why the authors rewrite all used methods for analysis. These methods are well known and must be cited. 11 honey samples are not sufficient for characterization of Algerian honey. Also most of the samples are polyfloral and one is monofloral Ziziphus lotus.
Author Response
We are grateful to you for insightful comments on our paper. We have been able to incorporate changes to reflect all the suggestions provided by the reviewers.
We improved the overall level of the manuscript according to your comments and suggestions, we also answered point by point. We hope that the result we will suit you.
Thank you for your suitable suggestion concerning our introduction, it was taken into consideration and the introduction was improved according to your comments. We added a part explaining why this study is important, as you suggested.

Round 2
Reviewer 1 Report
The manuscript has been thoroughly corrected, but the conclusions are still lacking. Reported a few minor errors.

Author Response
We would like to thank the reviewer #1 for his/her insightful and valuable comments on the manuscript. To address the concerns raised, the conclusion was improved, minor errors reported were corrected, and we want to draw your attention to the fact that we have used a paid English editing service. We also have carefully read the entire manuscript and removed a few minor typos.

Reviewer 2 Report
The text improved with the changes made.
Author Response
We would like to thank the reviewer #2 for his/her support of our work and insightful and valuable comments that helped us improve the manuscript.

This manuscript is a resubmission of an earlier submission. The following is a list of the peer review reports and author responses from that submission.